# The Significance of C-Reactive Protein Value and Tumor Grading for Malignant Tumors: A Systematic Review

**DOI:** 10.3390/diagnostics14182073

**Published:** 2024-09-19

**Authors:** Paul Șiancu, George-Călin Oprinca, Andra-Cecilia Vulcu, Monica Pătran, Adina Emilia Croitoru, Denisa Tănăsescu, Dan Bratu, Adrian Boicean, Ciprian Tănăsescu

**Affiliations:** 1Oncology Department, Sibiu County Emergency Clinical Hospital, 550245 Sibiu, Romania; pauls.siancu@ulbsibiu.ro (P.Ș.); mpatran@yahoo.com (M.P.); 2Preclinical Department, Faculty of Medicine, Lucian Blaga University of Sibiu, 550169 Sibiu, Romania; georgecalin.oprinca@ulbsibiu.ro; 3Pneumology Hospital Sibiu, 550196 Sibiu, Romania; vulcu.andra@yahoo.com; 4Oncology Department, Fundeni Clinical Institute, 022238 Bucharest, Romania; adina.croitoru09@yahoo.com; 5Medical Clinical Department, Faculty of Medicine, Lucian Blaga University of Sibiu, 550169 Sibiu, Romania; denisa.tanasescu@ulbsibiu.ro (D.T.); adrian.boicean@ulbsibiu.ro (A.B.); 6Surgical Clinical Department, Faculty of Medicine, Lucian Blaga University of Sibiu, 550169 Sibiu, Romania; tanasescuciprian@yahoo.fr; 7Surgical Department, Sibiu County Emergency Clinical Hospital, 550245 Sibiu, Romania; 8Gastroenterology Department, Sibiu County Emergency Clinical Hospital, 550245 Sibiu, Romania

**Keywords:** malignant tumors, c-reactive protein, histological differentiation, tumor grading, grading histology

## Abstract

Background: Malignant tumors represent a significant pathology with a profound global impact on the medical system. The fight against cancer represents a significant challenge, with multidisciplinary teams identifying numerous areas requiring improvement to enhance the prognosis. Facilitating the patient’s journey from diagnosis to treatment represents one such area of concern. One area of research interest is the use of various biomarkers to accurately predict the outcome of these patients. A substantial body of research has been conducted over the years examining the relationship between C-reactive protein (CRP) and malignant tumors. The existing literature suggests that combining imaging diagnostic modalities with biomarkers, such as CRP, may enhance diagnostic accuracy. Methods: A systematic review was conducted on the PubMed and Web of Science platforms with the objective of documenting the interrelationship between CRP value and tumor grading for malignant tumors. After the application of the exclusion and inclusion criteria, 17 studies were identified, published between 2002 and 2024, comprising a total of 9727 patients. Results: These studies indicate this interrelationship for soft tissue sarcomas and for renal, colorectal, esophageal, pancreatic, brain, bronchopulmonary, ovarian, and mesenchymal tumors. Conclusions: Elevated CRP levels are correlated with higher grading, thereby underscoring the potential utility of this biomarker in clinical prognostication.

## 1. Introduction

Cancer stands out as the most burdensome ailment in terms of its impact on health, society, and economic implications compared to all other human diseases. Its detrimental effects on individuals, both physically and socially, are quantified through the measurement of Disability-Adjusted Life Years (DALYs) attributed to the specific cause. The collective probability of developing cancer at any point between birth and the age of 74 is estimated at 20.2%. It is noteworthy that this probability exhibits slight gender-related variation, with males having a 22.4% probability of developing cancer during their lifetime, as compared to a 18.2% probability for females [1].

In the year 2020, it is estimated that there were approximately 19.3 million new cases of cancer worldwide. It should be noted that this figure excludes non-melanoma skin cancer. Of the total number of new cases of cancer, approximately 2.3 million were diagnosed as breast cancer, representing approximately 11.7% of all cancer diagnoses. This was followed by lung cancer at 11.4%, colorectal cancer at 10.0%, prostate cancer at 7.3%, and gastric cancer at 5.6% [2].

An accurate prognostic assessment can have a significant impact on the clinical management and outcomes of patients with malignant tumors. It facilitates the identification of the response to treatment of certain individuals and may also indicate resistance to chemotherapy. This translates into the ability to predict overall survival and progression-free survival in cancer patients. It is, therefore, of the utmost importance to identify reliable prognostic indicators [3].

CRP is an acute-phase reactant, produced by the liver as a result of various pro-inflammatory stimuli. It has been extensively employed as a marker for systematic inflammation in numerous medical conditions [4,5]. Recently, studies have indicated a correlation between elevated CRP levels and poor prognostic outcomes in cancer patients. Consequently, CRP has emerged as a potential prognostic biomarker for malignant diseases [6].

Malignant tumors constitute a group of diseases defined by the uncontrolled growth and invasion of abnormal cells. Histological examination plays a pivotal role in the diagnosis and comprehension of these tumors [7].

A histopathology report typically includes information on the tumor’s stage, histological type, and grade, which are determined by molecular and microscopic examination [8].

Grading categorization of the neoplasia can be defined based on the tumor’s microscopic appearance. Tumors are classified as low-grade or higher-grade based on the appearance of the cells and tissue structure [9].

In general, low-grade cancers are well differentiated, which indicates that the cells resemble healthy cellular counterparts. High-grade cancers, on the other hand, are anaplastic and are more clinically aggressive than low-grade cancers. The most poorly differentiated component of the tumor is the determining factor in the overall tumor grade. The grading systems categorize tumors into three or four grades based on cellular differentiation, with GX indicating that the grade cannot be evaluated, G1 in well-differentiated carcinomas, G2 in moderately differentiated carcinomas, and G3 and G4 in poorly differentiated or anaplastic neoplasms [10] (Figure 1).

The Nottingham grading system, recommended by the World Health Organization (WHO), is the preferred system for classifying invasive breast carcinoma based on the presence and degree of specific morphological alterations. These alterations include the extent of tubular formation, the degree of nuclear pleomorphism, and mitotic activity, with each feature assigned a score on a scale of 1 to 3. This generates a combined Nottingham score between 3 and 9, which is then used to determine the risk category: low-risk (3–4), intermediate-risk (5–6–7), or high-risk (8–9) carcinoma [11].

Studies that have shown certain links between blood CRP levels and specific cancer types—most frequently lung, breast, and colorectal cancer—can be found in the literature, generally with higher levels of CRP being associated with worse prognosis [12].

In the literature, there has been growing interest in the relationship between CRP values and the histological level of differentiation for various malignancies. While some studies have identified a strong correlation between the two, others have not [13,14,15].

The objective of this study was to collate and synthesize data from a range of specialized literature sources on the significance of the relationship between CRP and histopathology reports within primary tumor sites. Furthermore, this study aimed to identify and highlight specific tumor types that require further investigation in this particular area.

## 2. Materials and Methods

A comprehensive search was conducted on PubMed and Web of Science by the mesh phrases “c-reactive protein” and “tumor grading in malignant cancer” or “histopathology grade in malignant tumors” or “grade of differentiation in malignant cancer”. Included were all English-language articles for which the complete text could be found. Furthermore, a manual search was conducted in the references of the relevant review to gain additional insight into the subject matter. Book chapters, commentaries, editorials, letters, and meeting abstracts were excluded. The research was conducted by applying the PICOS strategy of research recommended by The Preferred Reporting Items for Systematic reviews and Meta-Analyses (PRISMA) guidelines [16], as follows: P: patients with confirmed malignant tumor; I: testing of the CRP value and tumor grading level; C: the CRP value compared to the tumor grading level; O: whether there is a statistically significant correlation between CRP and tumor grading; S: any types of clinical studies were included in the review.

In addition to confirming or invalidating the interrelationship between CRP and the degree of tumor differentiation, the present manuscript documents several other relevant details, including the number of patients enrolled, the year of publication of the study, protein values, and tumor grades.

A preliminary search of the literature identified 550 articles. Following the removal of duplicate entries and the application of established inclusion and exclusion criteria, 17 articles were selected for inclusion in the qualitative analysis. It is notable that all of the final articles included in this review were based on retrospective studies. To assess the quality of the studies, we applied the Newcastle–Ottawa Scale (NOS). The NOS is a tool utilized for the assessment of risk of bias in observational studies. The scale comprises three categories, namely, good quality, fair quality, and poor quality [17]. The studies included in this review are of high quality, with all studies scoring a minimum of seven points on the NOS. The research structure is illustrated in Figure 2.

The articles included in this review are all retrospective studies, published between 2002 and 2024, and encompass a total of 9727 patients. When we analyzed the literature, we wanted to check the following aspects of each study: the authors, the year of publication, the localization of the studied tumor, the histopathological type, the statistical significance of the researched interrelationship, the number of patients enrolled, the mean CRP value, and the predominant degree of differentiation in the studies (Table 1).

## 3. Results and Discussion

Out of the 17 studies included in this review, 16 indicate an association between CRP value and degree of tumor differentiation for malignant tumors. Thus, the interrelationship is found for renal tumors [18], colorectal cancer [19,20], colon cancer [21], esophageal tumors [22,23], pancreatic neuroendocrine tumors [24,25], pancreatic adenocarcinoma [26,27], soft tissue sarcoma [28,29], mesenchymal tumors [30], brain tumors [31], lung cancer [32], and ovarian cancer [14]. It is important to note that the paper by Huang W, Wu L, Liu X, et al. reports an association between CRP and tumor grading for esophageal squamous carcinoma. However, despite this, the statistical association is not sufficiently robust [22].

Scientific research has shown that the association could not be established for uterine malignancies [33].

Regarding colon and rectal tumors, we identified three studies that investigated the PCR–grading interrelationship, with all three confirming the correlation [19,20,21]. In the field of oncology, there is a significant challenge in understanding the differences in prognosis and therapeutic options between the right and left colon [34,35,36]. This topic has been extensively studied, yet there remains a lack of consensus. Publications have reported varying opinions on this matter. Bustamante-Lopez’s paper, for instance, states that the concept of treating the right and left colon as separate entities was proposed as early as 1990. The fundamental rationale behind this approach is the observation that they have undergone different embryological developments [34]. In their publication, Hodges and colleagues propose that colorectal cancer pathology should be viewed not as two distinct entities, the colon and rectum, but rather as three distinct entities: the right colon, the left colon, and the rectum [37]. Extremely intriguing is that the European Society for Medical Oncology (ESMO) guidelines, which dictate oncological practice across Europe, make a clear statement that the rectum is not a molecularly different entity. These guidelines recommend the use of anti-EGFR monoclonal antibodies for the left colon and anti-VEGF antibodies for the right colon [38]. The study by Plastiras and colleagues that addresses this debate demonstrates that, at comparable stages of disease, patients with left colon cancer have a superior prognosis compared to those with right colon tumors [35]. Bustamante-Lopez and his team reach the same conclusion [34].

Given the current status of this topic in the field of oncology, we believe that an additional step towards a deeper understanding of the observed differences in behavior would be to investigate the interrelationship between CRP and grading on separate lots, distributed as follows: right colon, left colon, and rectum. This would allow us to ascertain whether the observed differences are significant when considered individually for each lot. However, following a comprehensive literature review, we were unable to identify any studies that have previously addressed this question.

Furthermore, we wish to highlight the study conducted by Tolia and his team, which delves into the interrelationship regarding bronchopulmonary tumors. The study in question addresses non-small cell lung cancers in general, but does not specify the exact histopathology of the tumors under study, such as adenocarcinoma or squamous cell carcinoma [32]. As the tumor with the highest mortality rate among male cancer patients and exhibiting a wide range of histological subtypes (including adenocarcinoma, squamous cell carcinoma, large cell carcinoma, and other subsets) [39], we believe that studying the association according to each histopathological subtype represents an ongoing area of interest in oncology.

As for normal CRP reference values, these can vary from one laboratory to another, but the literature speaks of a normal range of between 0.8 mg/L and 3.0 mg/L [40]. It is also crucial to acknowledge that, when evaluating this acute-phase reactant, it is essential to consider the existence of multiple isoforms of CRP, each exhibiting distinct biological activities. The modified monomeric isoform (m-CRP) modulates inflammatory responses by binding to activated cell membranes, thereby stimulating platelets and leukocytes associated with acute phase responses and increasing the inflammatory process. It also ligates extracellular matrix components in the tissues involved. The pentameric isoform (p-CRP), which is the form quantified in PCR diagnostic measurements, is significantly less reactive, with poor anti-inflammatory bioactivity. Nevertheless, its accumulation in the bloodstream is linked to a sustained, low-grade inflammatory response, which suggests that the underlying disease process is ongoing and advancing, as observed in cancer [41].

In terms of sensitivity, CRP can be classified into two categories: conventional and highly sensitive. While both are employed for the detection of inflammation, high-sensitivity CRP is regarded as a marker of low-grade inflammation. Conventional CRP encompasses a broad range, making it a prevalent diagnostic tool for monitoring inflammation and early infection. However, its sensitivity is diminished at lower concentrations. High-sensitivity CRP testing can detect proteins at lower concentrations, and is therefore more effective in some situations than the conventional method [42].

The significance of high-sensitivity C-reactive protein levels in cancer is not yet known and has no established value [41].

Increased CRP indicates a tumor inflammatory microenvironment that is associated with tumor progression, but also with poor tumor differentiation [43].

Patients with cancer often exhibit diminished immune function, which can predispose them to infection. The most common infections among cancer patients are those of the respiratory, digestive, and urinary tracts. Such infections have the potential to elevate CRP levels [44].

A histopathological examination serves as the foundation for diagnosis and treatment guidance in malignant conditions. Despite its pivotal role in diagnosis, histopathological reports continue to present a range of challenges, resulting in delays in treatment. As identified by Mirham and colleagues, key challenges include the use of specialized terminology unique to the field of pathology, the implementation of diverse tumor grading systems, and the presence of vague expressions [45].

Any postponement in the administration of cancer-specific treatment has a detrimental effect on the survival of patients with cancer. Ward and colleagues’ research demonstrates the added relative risk of death increases by 7.4% with each deferral of treatment [46].

In the present era, a number of digital pathology software solutions, including artificial intelligence, are capable of studying the morphological characteristics of entire tissue slides and calculating a more concise grade based on cellular and tissue morphology. This approach effectively eliminates the potential for subjectivity inherent in the evaluation process conducted by a pathologist [47].

As evidenced in the literature, an accurate diagnosis and multidisciplinary collaboration have been demonstrated to have a profound impact on the management of cancer patients. This has been shown to improve overall survival and quality of life for these patients [48]. According to Pulumati and his team’s publication, an improvement in the accuracy and speed of the diagnostic process can be made by merging the information provided by the various diagnostic modalities, using data provided by imaging and molecular methods, with biomarkers consistently mentioned in this study [49]. CRP is considered such a biomarker, according to Nassar and his researchers’ publication [50].

### Limitations

Although the studies included in the review are comparable in terms of the research protocol, some differences were identified that may limit the generalizability of the findings. These include the lack of specification regarding the time at which the CRP was collected, the use of different isoforms of CRP and laboratory reference values, and the absence of mention of patient factors that could influence the CRP level, such as tumor stage, comorbidities, treatments, and infections.

## 4. Conclusions

A notable correlation was identified between the level of CRP and the degree of differentiation of malignant tumors.

While histopathological examination is the gold standard for diagnosing malignant tumors, it can still present challenges that impact the management of oncological patients. We propose addressing these issues through enhanced inter-clinical collaboration and integrating histopathological examination with specific biomarkers, such as CRP, as a potential solution.

## Figures and Tables

**Figure 1 diagnostics-14-02073-f001:**
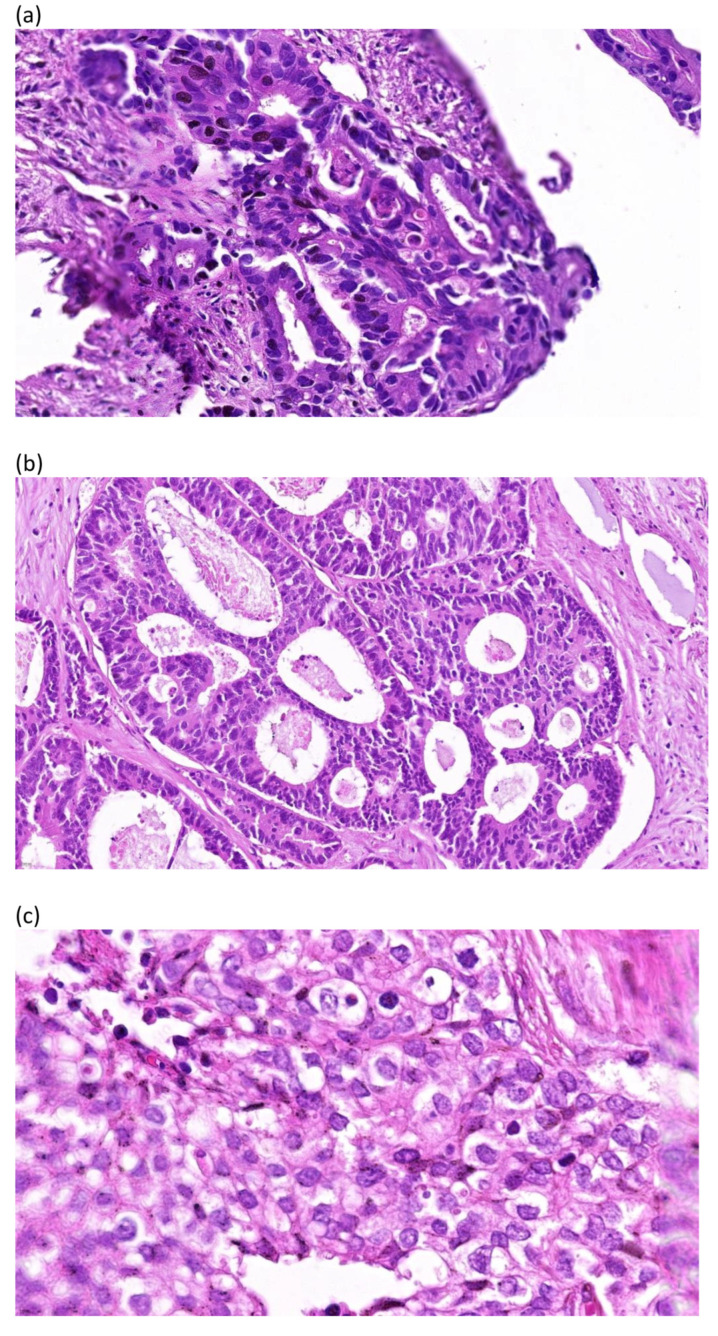
Degree of malignancy in neoplasms (colon adenocarcinoma). Personal case reports from Dr. George-Călin Oprinca. (**a**) Well-differentiated (G1) colon adenocarcinoma (Hematoxylin–Eosin staining—90.2×): Well-represented uniform gland formation, with simple tubules and round uniform, basal oriented nuclei. (**b**) Moderately differentiated (G2) colon adenocarcinoma (Hematoxylin–Eosin staining—50.4×): Conspicuous gland formation, represented by slightly irregular well-circumscribed gland formation, sometimes with cribriform pattern. Round or slightly irregular nuclei. (**c**) Poorly differentiated (G3) colon adenocarcinoma (Hematoxylin–Eosin staining—173.4×): Sheets of cells without gland formation composed by pleomorphic epithelioid cells and irregular nuclei.

**Figure 2 diagnostics-14-02073-f002:**
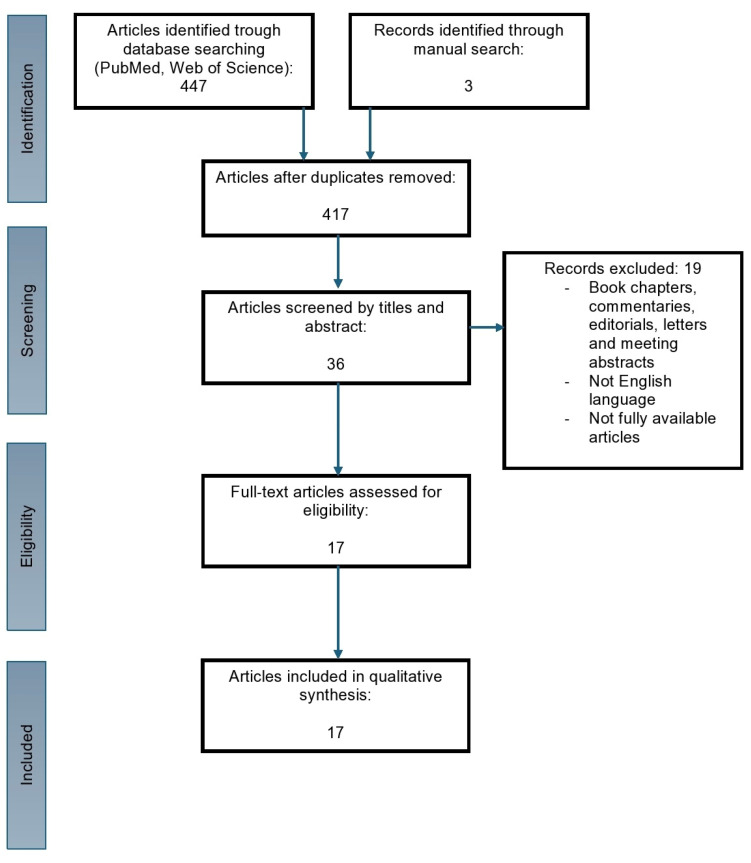
PRISMA flowchart for the articles included in the review. This process comprises four principal stages (introduction, screening, eligibility, and included), with each stage and criterion delineating the process by which the final number of included articles was determined.

**Table 1 diagnostics-14-02073-t001:** General characteristics of the articles included in this review. CRP: c-reactive protein, G1: grade 1, G2: grade 2, G3: grade 3.

Author, Year	Tumor Location	Histopathological Type	Statistical Significance	No. of Patients	Mean CRP Value	Predominant Grading
Hamidi N, Gökçe MI, Süer E, et al., 2015 [18]	Renal	Clear cell carcinoma	Yes, (*p* = 0.0001)	116	0.818 mg/L	G1–G2
Hidayat F, Labeda I, Sampetoding S, et al., 2021 [19]	Colorectal	Adenocarcinoma	Yes, (*p* = 0.005)	46	34.10 mg/L	G2
Huang L, Liu J, Huang X; et al., 2021 [20]	Colorectal	Adenocarcinoma	Yes, (*p* < 0.001)	2471	29.41 mg/L	G2
Kersten C, Louhimo J, Ålgars A; et al., 2013 [21]	Colon	Adenocarcinoma	Yes	525	>30 mg/L	G3
Huang W, Wu L, Liu X, et al., 2019 [22]	Esophagus	Squamous carcinoma	No, (*p* = 0.086)	961	18.92 mg/L	G2
Song ZB, Lin BC, Li B, et al., 2013 [23]	Esophagus	Squamous cell carcinoma	Yes, (*p* = 0.034)	156	<5 mg/L	G1/G2
Nießen A, Schimmack S, Sandini M, et al., 2021 [24]	Pancreas	Neuroendocrine tumor	Yes, (*p* < 0.001)	559	Low: <5 mg/L	G1
Primavesi F, Andreasi V, Hoogwater FJH, et al., 2020 [25]	Pancreas	Neuroendocrine tumor	Yes, (*p* = 0.004)	364	0.25 mg/L	G1
Szkandera J, Stotz M, Absenger G, et al., 2014 [26]	Pancreas	Adenocarcinoma	Yes, (*p* < 0.05)	474	23.2 mg/L	High grade
van Wijk L, de Klein GW, Kanters MA, et al., 2020 [27]	Pancreas	Adenocarcinoma	Yes (*p* < 0.001)	163	>1 mg/dL (>10 mg/L)	G3
Hashimoto K, Nishimura S, Shinyashiki Y, et al., 2022 [28]	Soft tissue of the torso and upper and lower limbs	Soft tissue sarcoma	Yes, (*p* = 0.008)	22	1.69 mg/L	G3
Szkandera J, Gerger A, Liegl-Atzwanger B, et al., 2013 [29]	Soft tissue	Sarcoma	Yes, (*p* < 0.05)	304	3.3 mg/L	G3
Nakanishi H, Araki N, Kudawara I, et al., 2002 [30]	Soft tissue—mesenchymal	Malignant fibrous histiocytoma	Yes, (*p* < 0.005)	46	3.7 mg/dL (37 mg/L)	G3
Strojnik T, Smigoc T, Lah TT; 2014 [31]	Brain	Glioma	Yes, (*p* = 0.02)	165	>5 mg/L	High grade
Tolia M, Tsoukalas N, Kyrgias G, et al., 2015 [32]	Non small-cell lung cancer	Not specified	Yes, (*p* < 0.001)	100	23.1 mg/L	G1/G2
Zhang W, Zhang Z, Qian L; 2024 [14]	Ovarian	Not specified	Yes, (*p* = 0.040)	3202	High	G3
Schwameis R, Grimm C, Petru E, et al., 2015 [33]	Uterus	Leiomyosarcoma	No, (*p* = 0.07)	53	3.46 mg/dL (34.6 mg/L)	G3

## Data Availability

The data presented in this study are available on request from the corresponding author. The data are not publicly available due to privacy reasons.

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
