# Peer review of "The Significance of C-Reactive Protein Value and Tumor Grading for Malignant Tumors: A Systematic Review"

_diagnostics, 2024, doi:10.3390/diagnostics14182073_

Round 1

Reviewer 1 Report

Comments and Suggestions for Authors

The review by Șiancu and co-authors is entitled "Understanding the Significance of C-Reactive Protein Value and Tumor Grading for Malignant Tumors: A Systematic Review".

In this manuscript the authors attempted to characterize the correlation between CRP levels and tumor grading. However, there are important issues that speak against its publication in the Diagnostics journal:

The text of the manuscript contains data (a number of Table 1 lines) where correlation between the CRP and the stage of the disease has been identified is in doubt; the “results and discussion” are essentially incomprehensible speculations in an attempt to justify the choice of a more than controversial topic for a review, where the authors try to generalize the material that is, in principle, non-generalizable (due to differences in CRP levels depending on tumor type; and that CRP levels have been shown to increase with obesity, smoking, postmenopausal hormone use, and to decrease with higher physical activity levels, better diet quality, and higher alcohol intake).

Table 1, Ref.#22: is p=0.086 statistically significant?

What were the reasons for providing the unpublished author's images in the review article?

Legends to figures: Can the authors, without reading the article itself, but only looking at the picture and legend to it, understand what the meaning is and what is the result? If not, then the legend is inappropriate.

The legends lack information about magnification (it differs within figures), staining etc.

The Abstract slightly exceeds the 200-word limit established in the journal's “Instructions for Authors” (204 words).

The Conclusions are not supported by the results.

References: do not meet the journal's “Instructions for Authors”.

Comments on the Quality of English Language

The language of the article does not meet publication standards. The authors must revise the manuscript using more refined academic language.

Author Response

Comments 1: The text of the manuscript contains data (a number of Table 1 lines) where correlation between the CRP and the stage of the disease has been identified is in doubt; the “results and discussion” are essentially incomprehensible speculations in an attempt to justify the choice of a more than controversial topic for a review, where the authors try to generalize the material that is, in principle, non-generalizable (due to differences in CRP levels depending on tumor type; and that CRP levels have been shown to increase with obesity, smoking, postmenopausal hormone use, and to decrease with higher physical activity levels, better diet quality, and higher alcohol intake)

Response 1: Dear Reviewer,

We would like to express our gratitude for taking the time to read our paper and for your valuable suggestions, which we believe will enhance the paper and, consequently, increase the likelihood of its publication. We have endeavored to adapt the paper in accordance with your recommendations.

Table 1 has been modified to eliminate any potential ambiguity. The results and discussion section is based on a comprehensive review of the literature, drawing on a range of authoritative sources and engaging with key debates in the field of medical oncology.

For further details on the limitations of this study, please refer to the section on limitations (lines 242-248). Additionally, please note the discussion on page 8, lines 217-220. Implicitly, there are numerous limitations to this approach due to the fact that there are several isoforms of CRP, the workup of biological samples in different laboratories, CRP collection at different times, and another limitation is represented by the factors of each patient that influence the CRP values.

Secondly, the numerous studies involving thousands of patients, documented over periods of decades, demonstrate that the topic underlying our work is of interest in the medical field and of contemporary interest that merits further investigation. Therefore, we believe that our review paper, which synthesizes the research conducted thus far, represents a significant advancement in the clarification of this topic, which some may consider "controversial."

Comments 2: Table 1, Ref.#22: is p=0.086 statistically significant?

Response 2: A very good observation. In Table 1, the original wording was modified to "not strong enough" in the statistical significance column in the case of the paper by Huang W, Wu L, Liu X, et al. (2019) [citation 22]. The authors proposed a correlation between CRP and tumor grading, yet the wording was confusing. In the results sentence, the authors combined both statistically significant and non-significant associations. The cited article states, "Before conducting PSM, the low-CRP group had a lower age (p = 0.001), lower histologic grade (p = 0.086), and lower TNM stage (p = 0.254)." Despite the authors' assertion that they have identified a correlation in the aforementioned proposition, the statistical p-value is not sufficiently robust.

Comments 3: What were the reasons for providing the unpublished author's images in the review article?

Response 3: The histopathologic examination is the most crucial diagnostic method in the field of oncology. We chose colon adenocarcinoma to illustrate the differentiation grades (G1, G2, G3) in our review because it is one of the most well-documented and studied types of cancer in terms of histological grading. The clear visual differences in the grades of colon adenocarcinoma make it an ideal candidate for demonstrating the concepts of tumor differentiation. Additionally, colon adenocarcinoma is prevalent and clinically significant.

Comments 4: Legends to figures: Can the authors, without reading the article itself, but only looking at the picture and legend to it, understand what the meaning is and what is the result? If not, then the legend is inappropriate.

Response 4: For Figure 1: The revised paper presents a single merged image. The figure caption has been expanded to include detailed information about the coloring and magnifications used. The legend reads as follows: Figure 1. Degree of malignancy in neoplasms (colon adenocarcinoma) Personal case reports dr. George-Călin Oprinca. a) Well differentiated (G1) colon adenocarcinoma (Hematoxylin-Eosin staining – 90.2x): Well represented uniform gland formation, with simple tubules and round uni-form, basal oriented nuclei; b) Moderately differentiated (G2) colon adenocarcinoma (Hematox-ylin-Eosin staining – 50.4x): Conspicuous gland formation, represented by slightly irregular well circumscribed gland formation, sometimes with cribriform pattern. Round or slightly irregular nuclei; c) Poorly differentiated (G3) colon adenocarcinoma (Hematoxylin-Eosin staining – 173.4x): Sheets of cells without gland formation composed by pleomorphic epithelioid cells and irregular nuclei. Consequently, figure number 4 has been redesignated as figure number 2 and we have augmented the legend of this figure: Figure 2. PRISMA flowchart for the articles included in the review. The PRISMA flow chart was utilized to select the articles to be included in this review. This process comprises four principal stages (introduction, screening, eligibility and included), with each stage and criterion delineating the process by which the final number of included articles was determined.

Comments 5: The legends lack information about magnification (it differs within figures), staining etc.

Response 5: The illustrated materials have been updated with additional description regarding staning and magnification: Figure 1. Degree of malignancy in neoplasms (colon adenocarcinoma) Personal case reports dr. George-Călin Oprinca. a) Well differentiated (G1) colon adenocarcinoma (Hematoxylin-Eosin staining – 90.2x): Well represented uniform gland formation, with simple tubules and round uniform, basal oriented nuclei; b) Moderately differentiated (G2) colon adenocarcinoma (Hematoxylin-Eosin staining – 50.4x): Conspicuous gland formation, represented by slightly irregular well circumscribed gland formation, sometimes with cribriform pattern. Round or slightly irregular nuclei; c) Poorly differentiated (G3) colon adenocarcinoma (Hematoxylin-Eosin staining – 173.4x): Sheets of cells without gland formation composed by pleomorphic epithelioid cells and irregular nuclei.

Comments 6: The Abstract slightly exceeds the 200-word limit established in the journal's “Instructions for Authors” (204 words)

Response 6: We are grateful for your observation. In order to comply with the 200-word limit, we have restructured the abstract: Malignant tumors represent a significant pathology with a profound global impact on the medical system. The fight against cancer represents a significant challenge, with multidisciplinary teams identifying numerous areas requiring improvement in order to enhance the prognosis. Facilitating the patient's journey from diagnosis to treatment represents one such area of concern. One area of research interest is the use of various biomarkers to accurately predict the outcome of these patients. A substantial body of research has been conducted over the years examining the relationship between C-reactive protein (CRP) and malignant tumors. The existing literature suggests that combining imaging diagnostic modalities with biomarkers, such as CRP, may enhance diagnostic accuracy. A systematic review was conducted on the PubMed and Web of Science platforms with the objective of documenting the interrelationship between CRP value and tumor grading for malignant tumors. After the application of the exclusion and inclusion criteria, 17 studies were identified, published between 2002 and 2024, comprising a total of 9,727 patients. These studies demonstrated this particular interrelationship for renal, colorectal, esophageal, pancreatic, brain, bronchopulmonary, ovarian, mesenchymal, and soft tissue sarcomas. Elevated CRP levels are correlated with higher grading, thereby underscoring the potential utility of this biomarker in clinical prognostication.  

Comments 7: The Conclusions are not supported by the results.

Response 7: Our first conclusion attests that interrelationship has been demonstrated for several localizations of malignant pathologies, please see lines 250 - 251. In the paper we extensively discuss the usefulness of confounding certain biomarkers, giving CRP as an example, with histopathologic findings and the importance of the multi-disciplinary approach to enhance. We respectfully invite you to re-read lines 55-72, 99-104 and 252-256.  

Comments 8: References: do not meet the journal's “Instructions for Authors”.

Response 8: For references we follow the journal’s “Instructions for Authors”. Our references are numbered in order of appearance in the text, are listed individually at the end of the manuscript, the numbers are placed in square brackets and before the punctuation. They also include the full title and are correctly cited.   Comments on the Quality of English Language: The language of the article does not meet publication standards. The authors must revise the manuscript using more refined academic language. Response on the Quality of English Language: We have refined the English wording in select sections of the paper. We kindly request that you re-examine the revised version of the article, in which we have highlighted in red the paragraphs that have been modified with the aim of achieving a more effective expression in English.

Reviewer 2 Report

Comments and Suggestions for Authors

This manuscript focuses on understanding the significance of C-reactive protein (CRP) value and the classification of malignant tumors. The study was carried out in accordance with Preferred Reporting Items for Systematic reviews and Meta-Analyses guidelines. In total 17 papers were included in the final qualitative analysis. The authors found that of the 17 studies selected, 16 studies demonstrated a statistically significant association between CRP value and degree of tumor differentiation. The manuscript is written in competent language, citing interesting facts and considerations regarding the prospects for using CRP as a cancer biomarker. I have two questions regarding the structure of the chapter Introduction:

Lines 89-94 – here is the text about breast cancer. However, the rest of the introduction talks about cancer in general, without specifying the type. Because of this paragraph, the logic of the narrative in the introduction chapter is disrupted. Explain why you chose to provide information about breast cancer in this paragraph.

Figures 1-3 do not indicate tumor type. Please, indicate it and explain in the text why did you decide to use this type of tumor to illustrate your text.

Author Response

Comments 1: Lines 89-94 – here is the text about breast cancer. However, the rest of the introduction talks about cancer in general, without specifying the type. Because of this paragraph, the logic of the narrative in the introduction chapter is disrupted. Explain why you chose to provide information about breast cancer in this paragraph.

Response 1: Dear Reviewer, We would like to express our gratitude for taking the time to read our paper and for your valuable suggestions, which we believe will enhance the paper and, consequently, increase the likelihood of its publication. We have endeavored to adapt the paper in accordance with your recommendations. Regarding the breast cancer paragraph, we aimed to describe a specific example where a scoring system, such as the Nottingham system, is used to assess tumor differentiation. This system evaluates several visual morphological changes, including the degree of tubular formation, nuclear pleomorphism, and mitotic activity. However, if the reviewer suggests that removing this paragraph would better align with the logic of the research, we are prepared to do so.

Comments 2: Figures 1-3 do not indicate tumor type. Please, indicate it and explain in the text why did you decide to use this type of tumor to illustrate your text.

Response 2: The histopathologic examination is the most crucial diagnostic method in the field of oncology. We chose colon adenocarcinoma to illustrate the differentiation grades (G1, G2, G3) in our review because it is one of the most well-documented and studied types of cancer in terms of histological grading. The clear visual differences in the grades of colon adenocarcinoma make it an ideal candidate for demonstrating the concepts of tumor differentiation. Additionally, colon adenocarcinoma is prevalent and clinically significant. Additionally, the paper presents a single merged image (figures 1, 2 and 3 merged into figure 1). The figure caption has been expanded to include detailed information about the coloring and magnifications used: Figure 1. Degree of malignancy in neoplasms (colon adenocarcinoma) Personal case reports dr. George-Călin Oprinca. a) Well differentiated (G1) colon adenocarcinoma (Hematoxylin-Eosin staining – 90.2x): Well represented uniform gland formation, with simple tubules and round uniform, basal oriented nuclei; b) Moderately differentiated (G2) colon adenocarcinoma (Hematoxylin-Eosin staining – 50.4x): Conspicuous gland formation, represented by slightly irregular well circumscribed gland formation, sometimes with cribriform pattern. Round or slightly irregular nuclei; c) Poorly differentiated (G3) colon adenocarcinoma (Hematoxylin-Eosin staining – 173.4x): Sheets of cells without gland formation composed by pleomorphic epithelioid cells and irregular nuclei.

Thank you for the constructive feedback.

Reviewer 3 Report

Comments and Suggestions for Authors

This manuscript reviews the levels of CRP value according to histological grade of cancer.  The study was well conducted and the conclusions valid; however, some points should be clarified:

1.      The title suggests that this review will describe the understanding of CRP in malignancy, which does not occur, the authors point out that the significance of CRP is not known (P7,p3,L192-193) the term should be discarded or the text should describe the “significance”.

2.      Figures 1-3 originate from the personal collection of one author (GCO), they could be either deleted or described in a single figure as degrees of malignancy in neoplasms.

3.     Table 1 displays the reviewed studies which are already placed in the references, the only important datum of this table is the summary of 9727 cases (P4,p4,L132-133) the contents and ciphers of Table 1 could be briefly synthesized in the text.

Author Response

Comments 1: The title suggests that this review will describe the understanding of CRP in malignancy, which does not occur, the authors point out that the significance of CRP is not known (P7,p3,L192-193) the term should be discarded or the text should describe the “significance”.

Response 1: Dear Reviewer, we would like to express our gratitude for taking the time to read our paper and for your valuable suggestions, which we believe will enhance the paper and, consequently, increase the likelihood of its publication. We have endeavored to adapt the paper in accordance with your recommendations. We acknowledge and appreciate your concerns regarding the selection of appropriate terminology in the title. In light of these concerns, we have opted to remove the term "understanding" from the title. The revised title is as follows: “The significance of C-reactive protein value and tumor grading for malignant tumors: A systematic review”.

Comments 2: Figures 1-3 originate from the personal collection of one author (GCO), they could be either deleted or described in a single figure as degrees of malignancy in neoplasms.

Response 2: We continue to believe that images of histopathologic examinations add value to our work, and therefore recommend their inclusion in the article. Going forward the revised paper presents a single merged image (figures 1, 2 and 3 merged into figure 1). The figure caption has been expanded to include detailed information about the coloring and magnifications used: Figure 1. Degree of malignancy in neoplasms (colon adenocarcinoma) Personal case reports dr. George-Călin Oprinca. a) Well differentiated (G1) colon adenocarcinoma (Hematoxylin-Eosin staining – 90.2x): Well represented uniform gland formation, with simple tubules and round uniform, basal oriented nuclei; b) Moderately differentiated (G2) colon adenocarcinoma (Hematoxylin-Eosin staining – 50.4x): Conspicuous gland formation, represented by slightly irregular well circumscribed gland formation, sometimes with cribriform pattern. Round or slightly irregular nuclei; c) Poorly differentiated (G3) colon adenocarcinoma (Hematoxylin-Eosin staining – 173.4x): Sheets of cells without gland formation composed by pleomorphic epithelioid cells and irregular nuclei.

Comments 3: Table 1 displays the reviewed studies which are already placed in the references, the only important datum of this table is the summary of 9727 cases (P4,p4,L132-133) the contents and ciphers of Table 1 could be briefly synthesized in the text.

Response 3: The structure of the table follows for each article incorporated into our study the following elements: the authors, the year of publication, the localization of the studied tumor, the histopathological type, the statistical significance of the researched interrelationship, the number of patients enrolled, the mean CRP value, and the predominant degree of differentiation in the studies. We briefly synthesized this into the revised text: The articles included in this review are all retrospective studies and were published between 2002 and 2024, with a total number of reported patients of 9727. When we analyzed the literature, we wanted to check the following aspects of each study: the authors, the year of publication, the localization of the studied tumor, the histopathological type, the statistical significance of the researched interrelationship, the number of patients enrolled, the mean CRP value, and the predominant degree of differentiation in the studies

(Table 1).

Round 2

Reviewer 1 Report

Comments and Suggestions for Authors

Unfortunately, the authors have addressed my concerns just partially. The reviewer finds the improvements of the manuscript made in the revised version to be insufficient to approve the current paper for publication. The reasons are listed below:

Comments 1: The text of the manuscript contains data (a number of Table 1 lines) where correlation between the CRP and the stage of the disease has been identified is in doubt; the “results and discussion” are essentially incomprehensible speculations in an attempt to justify the choice of a more than controversial topic for a review, where the authors try to generalize the material that is, in principle, non-generalizable (due to differences in CRP levels depending on tumor type; and that CRP levels have been shown to increase with obesity, smoking, postmenopausal hormone use, and to decrease with higher physical activity levels, better diet quality, and higher alcohol intake)

Response 1: Dear Reviewer,

We would like to express our gratitude for taking the time to read our paper and for your valuable suggestions, which we believe will enhance the paper and, consequently, increase the likelihood of its publication. We have endeavored to adapt the paper in accordance with your recommendations.

Table 1 has been modified to eliminate any potential ambiguity. The results and discussion section is based on a comprehensive review of the literature, drawing on a range of authoritative sources and engaging with key debates in the field of medical oncology.

For further details on the limitations of this study, please refer to the section on limitations (lines 242-248). Additionally, please note the discussion on page 8, lines 217-220. Implicitly, there are numerous limitations to this approach due to the fact that there are several isoforms of CRP, the workup of biological samples in different laboratories, CRP collection at different times, and another limitation is represented by the factors of each patient that influence the CRP values.

Secondly, the numerous studies involving thousands of patients, documented over periods of decades, demonstrate that the topic underlying our work is of interest in the medical field and of contemporary interest that merits further investigation. Therefore, we believe that our review paper, which synthesizes the research conducted thus far, represents a significant advancement in the clarification of this topic, which some may consider "controversial."

Reviewer's comments Round 2 #1 (RCR2 #1):

The Table 1 still remains controversial. CRP levels corresponding to G1/G2/G3 seem to depend on tumor type or any other factors: 1.69 mg/l for soft tissue sarcoma is G3, whereas 23.1 mg/L for NSCLC is G1/G2, so these data contradict your conclusion that "a notable correlation was identified between the level of CRP and the degree of differentiation of malignant tumors".

"High" for ovarian carcinoma: how much is it in milligrams per liter?

Comments 2: Table 1, Ref.#22: is p=0.086 statistically significant?

Response 2: A very good observation. In Table 1, the original wording was modified to "not strong enough" in the statistical significance column in the case of the paper by Huang W, Wu L, Liu X, et al. (2019) [citation 22]. The authors proposed a correlation between CRP and tumor grading, yet the wording was confusing. In the results sentence, the authors combined both statistically significant and non-significant associations. The cited article states, "Before conducting PSM, the low-CRP group had a lower age (p = 0.001), lower histologic grade (p = 0.086), and lower TNM stage (p = 0.254)." Despite the authors' assertion that they have identified a correlation in the aforementioned proposition, the statistical p-value is not sufficiently robust.

RCR2 #2:

If p=0.086 is " not strong enough", then p=0.07 is "not significant". Another brick in the wall of Table 1 controversy. And so on.

Comments 3: What were the reasons for providing the unpublished author's images in the review article?

Response 3: The histopathologic examination is the most crucial diagnostic method in the field of oncology. We chose colon adenocarcinoma to illustrate the differentiation grades (G1, G2, G3) in our review because it is one of the most well-documented and studied types of cancer in terms of histological grading. The clear visual differences in the grades of colon adenocarcinoma make it an ideal candidate for demonstrating the concepts of tumor differentiation. Additionally, colon adenocarcinoma is prevalent and clinically significant.

RCR2 #3:

OK. Approved.

Comments 4: Legends to figures: Can the authors, without reading the article itself, but only looking at the picture and legend to it, understand what the meaning is and what is the result? If not, then the legend is inappropriate.

Response 4: For Figure 1: The revised paper presents a single merged image. The figure caption has been expanded to include detailed information about the coloring and magnifications used. The legend reads as follows: Figure 1. Degree of malignancy in neoplasms (colon adenocarcinoma) Personal case reports dr. George-Călin Oprinca. a) Well differentiated (G1) colon adenocarcinoma (Hematoxylin-Eosin staining – 90.2x): Well represented uniform gland formation, with simple tubules and round uni-form, basal oriented nuclei; b) Moderately differentiated (G2) colon adenocarcinoma (Hematox-ylin-Eosin staining – 50.4x): Conspicuous gland formation, represented by slightly irregular well circumscribed gland formation, sometimes with cribriform pattern. Round or slightly irregular nuclei; c) Poorly differentiated (G3) colon adenocarcinoma (Hematoxylin-Eosin staining – 173.4x): Sheets of cells without gland formation composed by pleomorphic epithelioid cells and irregular nuclei. Consequently, figure number 4 has been redesignated as figure number 2 and we have augmented the legend of this figure: Figure 2. PRISMA flowchart for the articles included in the review. The PRISMA flow chart was utilized to select the articles to be included in this review. This process comprises four principal stages (introduction, screening, eligibility and included), with each stage and criterion delineating the process by which the final number of included articles was determined.

RCR2 #4:

OK.

Comments 5: The legends lack information about magnification (it differs within figures), staining etc.

Response 5: The illustrated materials have been updated with additional description regarding staning and magnification: Figure 1. Degree of malignancy in neoplasms (colon adenocarcinoma) Personal case reports dr. George-Călin Oprinca. a) Well differentiated (G1) colon adenocarcinoma (Hematoxylin-Eosin staining – 90.2x): Well represented uniform gland formation, with simple tubules and round uniform, basal oriented nuclei; b) Moderately differentiated (G2) colon adenocarcinoma (Hematoxylin-Eosin staining – 50.4x): Conspicuous gland formation, represented by slightly irregular well circumscribed gland formation, sometimes with cribriform pattern. Round or slightly irregular nuclei; c) Poorly differentiated (G3) colon adenocarcinoma (Hematoxylin-Eosin staining – 173.4x): Sheets of cells without gland formation composed by pleomorphic epithelioid cells and irregular nuclei.

RCR2 #5:

The images in Figure 1 should be presented with the same magnification level (f.ex. 50x), however if the authors need to provide a more magnified image (f.ex., 173x), they can zoom the appropriate image fragments in (near the 50x magnified picture).

Comments 6: The Abstract slightly exceeds the 200-word limit established in the journal's “Instructions for Authors” (204 words)

Response 6: We are grateful for your observation. In order to comply with the 200-word limit, we have restructured the abstract: Malignant tumors represent a significant pathology with a profound global impact on the medical system. The fight against cancer represents a significant challenge, with multidisciplinary teams identifying numerous areas requiring improvement in order to enhance the prognosis. Facilitating the patient's journey from diagnosis to treatment represents one such area of concern. One area of research interest is the use of various biomarkers to accurately predict the outcome of these patients. A substantial body of research has been conducted over the years examining the relationship between C-reactive protein (CRP) and malignant tumors. The existing literature suggests that combining imaging diagnostic modalities with biomarkers, such as CRP, may enhance diagnostic accuracy. A systematic review was conducted on the PubMed and Web of Science platforms with the objective of documenting the interrelationship between CRP value and tumor grading for malignant tumors. After the application of the exclusion and inclusion criteria, 17 studies were identified, published between 2002 and 2024, comprising a total of 9,727 patients. These studies demonstrated this particular interrelationship for renal, colorectal, esophageal, pancreatic, brain, bronchopulmonary, ovarian, mesenchymal, and soft tissue sarcomas. Elevated CRP levels are correlated with higher grading, thereby underscoring the potential utility of this biomarker in clinical prognostication.

RCR2 #6:

The Abstract was improved, but the fragment above marked in bold should be changed. You've studied not only sarcomae, but a number of other tumor types.

Comments 7: The Conclusions are not supported by the results.

Response 7: Our first conclusion attests that interrelationship has been demonstrated for several localizations of malignant pathologies, please see lines 250 - 251. In the paper we extensively discuss the usefulness of confounding certain biomarkers, giving CRP as an example, with histopathologic findings and the importance of the multi-disciplinary approach to enhance. We respectfully invite you to re-read lines 55-72, 99-104 and 252-256.

RCR2 #7:

Your conclusions are based on the data from very controversial and speculative Table 1.

Comments 8: References: do not meet the journal's “Instructions for Authors”.

Response 8: For references we follow the journal’s “Instructions for Authors”. Our references are numbered in order of appearance in the text, are listed individually at the end of the manuscript, the numbers are placed in square brackets and before the punctuation. They also include the full title and are correctly cited. Comments on the Quality of English Language: The language of the article does not meet publication standards. The authors must revise the manuscript using more refined academic language. Response on the Quality of English Language: We have refined the English wording in select sections of the paper. We kindly request that you re-examine the revised version of the article, in which we have highlighted in red the paragraphs that have been modified with the aim of achieving a more effective expression in English.

RCR2 #8:

References: still do not meet the journal's “Instructions for Authors”.

Please compare the correct format of references:

DeSantis, C.E.; Ma, J.; Gaudet, M.M.; Newman, L.A.; Miller, K.D., Sauer, A.G., Jemal, A.; Siegel, R.L. Breast cancer statistics. CA- Cancer J Clin 2019, 69, 438–451. doi: 10.3322/caac.21583.

with your style:

Zhu M, Ma Z, Zhang X, Hang D, Yin R, Feng J, et al. C-reactive protein and cancer risk: a pan-cancer study of prospective cohort and Mendelian randomization analysis. BMC Med. 2022 Sep 19;20(1):301.

Is it clear now?

Some references are irrelevant to text referring to them (f.ex., lines 55-60 and Ref.#3).

The English language was, indeed, refined if compared with previous manuscript version, but still contains a number of mistakes and poor style.

Comments on the Quality of English Language

English is better than in previous version, but still not perfect.

Author Response

Comments 1: Unfortunately, the authors have addressed my concerns just partially. The reviewer finds the improvements of the manuscript made in the revised version to be insufficient to approve the current paper for publication. The reasons are listed below: 

The Table 1 still remains controversial. CRP levels corresponding to G1/G2/G3 seem to depend on tumor type or any other factors: 1.69 mg/l for soft tissue sarcoma is G3, whereas 23.1 mg/L for NSCLC is G1/G2, so these data contradict your conclusion that "a notable correlation was identified between the level of CRP and the degree of differentiation of malignant tumors".

"High" for ovarian carcinoma: how much is it in milligrams per liter?

Response 1: Dear Reviewer, 

We would like to extend our gratitude to you once more for dedicating your time and attention to a second review of our paper. Your contribution is highly valued.
We would like to acknowledge that we did indeed respond to all of your comments from the initial round. We proceeded in a methodical manner, indicating any changes made based on your proposals. In instances where we deemed that a modification was not necessary, we provided a rationale argument and directed you to the specific text passages that supported our original formulation.

It is widely recognized within the oncological community that tumors and types of cancer exhibit significant heterogeneity. This is the reason why different stages and therapies are employed for each type of tumor. It is therefore reasonable to conclude that tumors of different types and locations will have varying effects on the host immune response. From an oncological perspective, this is not a controversial finding; it is to be expected. 
It is for this reason that we have included in our review table the tumor location, the histopathological type, the CPR value and the grading of the tumor. We do not perceive any contradiction; rather, we observe that tumors behave differently depending on their histological type and localization. 
The data presented in our table is the available data that was published in the papers cited.
We acknowledge your assertion that this topic is controversial. However, we believe that the publication of our paper should encourage further discussion and contribute to a deeper understanding of the significance of this correlation, which will ultimately reduce the potential for further controversy.

Comments 2: If p=0.086 is " not strong enough", then p=0.07 is "not significant". Another brick in the wall of Table 1 controversy. And so on.

Response 2: We have amended the wording to indicate that the results are not statistically significant. Despite this, the cited article does indicate a correlation between the variables, albeit not to a statistically significant degree. It should be noted that this is merely a citation of data that has been previously published in scientific literature; it does not represent our own original findings.

Comments 3: OK. Approved.

Response 3: We would like to express our sincerest gratitude for your invaluable contribution, which has significantly enhanced the quality of our paper.

Comments 4: OK.

Response 4: We would like to express our gratitude once more for your assistance.

Comments 5: The images in Figure 1 should be presented with the same magnification level (f.ex. 50x), however if the authors need to provide a more magnified image (f.ex., 173x), they can zoom the appropriate image fragments in (near the 50x magnified picture).

Response 5: In light of the fact that we presented the microscopic images regarding the grade of differentiation, the pathologist who digitalized the HE slides, who is also an expert in digital pathology, selected the optimal magnification for each image so that the images would display all histopathological characteristics of the presented grade (G1, G2, G3). It is our opinion that modifying the magnification of the images and then loading them with corner magnifications is an inappropriate approach, as it will undoubtedly lead to confusion among readers. The majority of published articles in the literature include microscopic images at varying magnifications. It is the responsibility of the pathologist to identify the optimal magnification for accurately depicting the described microscopic change.

Comments 6: The Abstract was improved, but the fragment above marked in bold should be changed. You've studied not only sarcomae, but a number of other tumor types.

Response 6: We would like to thank the reviewer for pointing out that the abstract has undergone improvements. We appreciate the initial suggestion. It is worth noting that our paper did not state that the focus was exclusively on sarcomas. However, we recognize that the previous wording could have caused confusion. Therefore, we have made changes to clarify this point.

Comments 7: Your conclusions are based on the data from very controversial and speculative Table 1.

Response 7: It is worth reiterating that the data presented in Table 1 is not our original findings but drawn from published literature. Our conclusions are based on a rigorous review of the literature and are duly referenced.

Comments 8: References: still do not meet the journal's “Instructions for Authors”.

Please compare the correct format of references:

DeSantis, C.E.; Ma, J.; Gaudet, M.M.; Newman, L.A.; Miller, K.D., Sauer, A.G., Jemal, A.; Siegel, R.L. Breast cancer statistics. CA- Cancer J Clin 2019, 69, 438–451. doi: 10.3322/caac.21583.

with your style:

Zhu M, Ma Z, Zhang X, Hang D, Yin R, Feng J, et al. C-reactive protein and cancer risk: a pan-cancer study of prospective cohort and Mendelian randomization analysis. BMC Med. 2022 Sep 19;20(1):301.

Is it clear now?

Some references are irrelevant to text referring to them (f.ex., lines 55-60 and Ref.#3).

The English language was, indeed, refined if compared with previous manuscript version, but still contains a number of mistakes and poor style.

Response 8: It is clear now, thank you. The requisite alterations have been implemented in accordance with your specifications. Furthermore, an additional examination of lines 55-60 has been conducted. The English language has undergone another revision.

Round 3

Reviewer 1 Report

Comments and Suggestions for Authors

The authors have addressed most of my concerns. The reviewer finds the revised version of the manuscript much improved in term of data presentation and organization and approves it for the publication in the Diagnostics journal after bringing the bibliography in accordance with the journal's requirements (references still do not meet the journal's “Instructions for Authors”).

Comments on the Quality of English Language

The reviewer finds the English language quality to be almost suitable. Just moderate editing is required.